# Assessment of Cyclin D1 Expression: Prognostic Value and Functional Insights in Endometrial Cancer: In Silico Study

**DOI:** 10.3390/ijms26030890

**Published:** 2025-01-22

**Authors:** Marcin Szymański, Dominika Jerka, Klaudia Bonowicz, Paulina Antosik, Maciej Gagat

**Affiliations:** 1Department of Histology and Embryology, Collegium Medicum in Bydgoszcz, Nicolaus Copernicus University in Torun, 85-092 Bydgoszcz, Poland; md.marcinszymanski@gmail.com (M.S.); dominika.jerka@cm.umk.pl (D.J.); klaudia.bonowicz@cm.umk.pl (K.B.); 2Faculty of Medicine, Collegium Medicum, Mazovian Academy in Płock, 09-402 Płock, Poland; 3Department of Clinical Pathomorphology, Faculty of Medicine, Collegium Medicum in Bydgoszcz, Nicolaus Copernicus University in Toruń, 85-094 Bydgoszcz, Poland; paulina.antosik@cm.umk.pl

**Keywords:** CCND1, endometrial cancer, prognostic factor

## Abstract

Endometrial cancer (EC) is the sixth most common cancer in women worldwide, with rising incidence, particularly in economically developed countries where obesity and type 2 diabetes are prevalent risk factors. EC comprises various histological subtypes with distinct behaviors: Type I tumors are generally estrogen-driven with favorable prognosis, while Type II tumors are hormone-independent, aggressive, and associated with poorer outcomes. Dysregulation of the cell cycle, particularly through cyclin-dependent kinases (CDKs) and their regulators like Cyclin D1 (CCND1), plays a crucial role in EC progression and recurrence. Cyclin D1 overexpression is often observed in the early stages of endometrioid carcinoma and complex hyperplasia, marking potential early carcinogenic events, while lower expression levels are common in high-grade subtypes like serous carcinoma. Although CDK inhibitors targeting Cyclin D1/CDK4/6 complexes have shown therapeutic potential in cancers such as breast and lung, their role in EC remains underexplored. This study integrates immunohistochemical evaluations of Cyclin D1 expression in EC patient samples with data from The Cancer Genome Atlas (TCGA) to assess its prognostic significance across EC subtypes. By correlating molecular, histopathological, and clinical outcomes, we aim to clarify the impact of Cyclin D1 dysregulation on EC progression and recurrence. Our findings may inform more personalized therapeutic approaches, particularly for high-grade and treatment-resistant forms of EC.

## 1. Introduction

EC ranks as the sixth most common cancer in women globally. In 2019 alone, 435,041 new cases were recorded, along with 91,641 related deaths worldwide [1]. In 2020, there were approximately 417,000 new cases, and this number has risen sharply over the past two decades [2]. With endometrial cancer often diagnosed around the seventh decade of life, many women may live another 10 to 25 years post-treatment, making its impact on their long-term quality of life a crucial consideration [3]. The rise in endometrial cancer is most notable in economically developed countries, where obesity and type 2 diabetes—key factors linked to its development—have become more prevalent. Encouragingly, most diagnoses occur when the disease is still localized; around 20% involve regional spread, while only 9% present with distant metastases [4]. Notably, the 5-year relative survival rate for EC is 81% overall but differs significantly across racial groups—84% for white women versus 63% for black women [5]. EC originates from the inner epithelial lining of the uterus, exhibiting diverse histological subtypes and molecular phenotypes. Traditionally, it has been classified into two main types based on hormonal influences and cellular characteristics. Type I EC is associated with unopposed estrogen exposure, typically involving low-grade, well-differentiated endometrioid adenocarcinomas (grades I and II) that are more common and associated with a favorable prognosis. Conversely, Type II EC is not estrogen-driven and is generally more aggressive, encompassing high-grade tumors such as grade III endometrioid adenocarcinomas, serous clear cell, undifferentiated carcinomas, and carcinosarcomas, which are less common but linked to a poorer prognosis [6].

EC progression and recurrence are closely linked to dysregulation in the cell cycle, which is orchestrated by CDKs and their regulatory cyclins. Among these, Cyclin D1 and its associated CDK4/6 complexes play a crucial role in driving cells through the G1 phase of the cell cycle, facilitating the transition to the S phase and, subsequently, DNA replication [7]. Aberrant overexpression of Cyclin D1 has been observed in various cancers, including endometrial cancer, and is often associated with unchecked cellular proliferation, tumor progression, and poor prognosis [8]. In EC specifically, Cyclin D1 overexpression may represent an early event in carcinogenesis, often marking the transition from normal endometrial tissue through hyperplastic stages to carcinoma. Studies have shown that Cyclin D1 is particularly overexpressed in complex hyperplasia and endometrioid carcinoma, whereas lower expression is more commonly associated with high-grade subtypes like serous carcinoma, which may follow alternative oncogenic pathways [9]. In recent years, the therapeutic potential of CDK inhibitors has been explored across several cancer types, such as breast and lung cancers, with promising results, particularly in targeting advanced or resistant forms [10]. However, the role of CDK dysregulation in EC remains understudied despite evidence linking altered Cyclin D1/CDK4/6 activity to EC cell proliferation and survival [11]. Understanding the precise role of Cyclin D1 and CDK regulation in different subtypes of EC is essential, as it could uncover new treatment strategies, especially for high-grade and recurrent cases that are resistant to conventional therapies [12].

In this study, we aim to integrate immunohistochemical evaluations of Cyclin D1 expression in EC patients with data from the Cancer Genome Atlas (TCGA) to provide a comprehensive analysis of its prognostic value across different EC subtypes. By comparing patient outcomes with molecular and histopathological data, this research seeks to clarify the role of Cyclin D1 dysregulation in EC progression and recurrence, potentially guiding more personalized therapeutic approaches.

## 2. Results

### 2.1. *CCND1* Immunoexpression in Endometrial Cancer and Adjacent Normal Tissue

CCND1 staining demonstrated nuclear localization in both normal and cancerous endometrial tissues. Negative control staining in normal endometrial tissue showed no detectable expression (Figure 1A), confirming the absence of nonspecific antibody binding. In endometrioid carcinoma, CCND1 expression ranged from low to strong, as illustrated in (Figure 1B) and (Figure 1C), respectively, demonstrating variability in expression within this histological subtype. In serous carcinoma, high nuclear expression of CCND1 was detected in several samples (Figure 1D), while other serous carcinoma cases showed low CCND1 expression (Figure 1E). These images reflect distinct expression patterns observed in tumor samples, indicating varying levels of CCND1 in different histological subtypes of endometrial cancer.

When analyzed as continuous variables, there was no significant difference in CCND1 staining intensity between the tumor and adjacent normal tissues for endometrial carcinoma overall (*p* = 0.48, Mann–Whitney test; Figure 2A). Similarly, for endometrioid adenocarcinoma, no statistically significant change was observed between the tumor and adjacent normal tissues (*p* = 0.66, Mann–Whitney test; Figure 2B). In non-endometrioid adenocarcinoma, CCND1 staining intensity also showed no significant difference between the tumor and adjacent tissues (*p* = 0.2, Mann–Whitney test; Figure 2C). Descriptive statistics for CCND1 staining across these subgroups are summarized in Table 1.

When the established definition of low and high expression for each category was applied, 98 (76.56%) and 30 (23.44%) cases demonstrated high- and low-intensity staining tumors for CCND1, respectively. When the percentage of staining was scored, 13 (10.16%) cases exhibited a high percentage of cell staining for CCND1, while the remaining 115 (89.84%) cases showed a low percentage of staining. Based on the formula for assessing the total immunoscore, 11 (8.59%) cases had tumors that presented CCND1 overexpression, while the remaining 117 (91.4%) cases displayed underexpression of this protein.

### 2.2. Tumor Characteristics with Respect to *CCND1* Immunoexpression

We evaluated whether CCND1 reactivity varied based on the clinicopathological features of the patients. When the established definition of low and high expression for each category was applied, 98 (76.56%) patients demonstrated high CCND1 expression, while 30 (23.44%) showed low expression. Among patients aged ≤ 60 years, 31 (75.61%) exhibited high CCND1 expression, compared to 67 (77.01%) in patients aged > 60 years. Regarding histological grade, 5 (55.56%), 59 (80.82%), and 34 (73.91%) patients with G1, G2, and G3 tumors, respectively, demonstrated high CCND1 expression. For pT status, high expression was observed in 35 (75.77%), 29 (80.56%), and 13 (76.47%) patients with T1, T2, and T3 + T4 tumors, respectively. Among histological types, 79 (77.45%) endometrioid cancers and 19 (73.08%) non-endometrioid cancers exhibited high CCND1 expression. Overall, while high CCND1 expression was prevalent across the categories, the analysis showed no significant associations between CCND1 expression and variables such as age, histologic grade, pT status, pN status, pM status, FIGO stage, LVSI, or histological type (*p* = 0.05, Fisher’s exact test; Table 1). 

### 2.3. Analysis of *CCND1* Protein Expression in Relation to Overall Survival

Kaplan–Meier survival analysis revealed that CCND1 immunohistochemical expression, defined using a cut-off value of 1, demonstrated a potential association with OS in endometrial cancer patients. Patients with CCND1 expression levels exceeding this threshold were classified as having high expression, whereas those with levels equal to or below 1 were categorized as having low expression. The median survival time for patients with high and low levels of CCND1 expression was 106.5 days. Among all endometrial cancer cases, high CCND1 expression exhibited a trend toward poorer OS compared to low expression; however, this difference was not statistically significant (*p* = 0.12; Figure 3A). In the endometrioid subtype, CCND1 expression levels showed a non-significant trend toward poorer OS with high expression (*p* = 0.31; Figure 3B). Similarly, within the non-endometrioid subtype, high CCND1 expression was associated with poorer survival outcomes, but this trend was also not statistically significant (*p* = 0.22; Figure 3C). In all cases, high CCND1 expression correlated with worse survival times, but none of the associations reached statistical significance.

### 2.4. *CCND1* mRNA Expression in Endometrial Cancer and Normal Tissue Based on TGCA Databases

Next, we analyzed publicly available expression data from TCGA for normal endometrial tissue and endometrial cancer. The analysis revealed that CCND1 mRNA expression levels were significantly upregulated in endometrial cancer compared to normal endometrial tissues (*p*-values shown in Figure 4**,** Mann–Whitney test). High expression levels were observed in endometrioid adenocarcinoma (EAC) and non-endometrioid adenocarcinoma (nEAC) subtypes, with distinct patterns between these groups.

### 2.5. Functional Enrichment Analysis

The top 50 genes positively correlated with CCND1 were identified using the TCGA dataset through the cBioPortal platform. Correlation analyses were performed using Spearman’s rank correlation. The highest positive correlations were observed for SPRY2 (ρ = 0.519, *p* = 1.08 × 10^−37^), SPRY4 (ρ = 0.465, *p* = 1.27 × 10^−29^), ETV4 (ρ = 0.440, *p* = 2.59 × 10^−26^), SPRY1 (ρ = 0.420, *p* = 6.26 × 10^−24^), and ETV5 (ρ = 0.418, *p* = 1.00 × 10^−23^) (Table 2). All listed genes demonstrated moderate positive correlations, with *p*-values indicating strong statistical associations.

Similarly, the analysis identified the top 50 genes negatively correlated with CCND1 using the TCGA dataset through the cBioPortal platform. Negative correlations were evaluated using Spearman’s rank correlation, as with the positively correlated genes. The most pronounced negative correlations were observed for CDKN2B-AS1 (ρ = −0.400, *p* = 1.20 × 10^−21^), MCCC1 (ρ = −0.354, *p* = 5.83 × 10^−17^), PTGS1 (ρ = −0.352, *p* = 8.03 × 10^−17^), CEP70 (ρ = −0.346, *p* = 2.67 × 10^−16^), and SVOPL (ρ = −0.346, *p* = 3.03 × 10^−16^) (Table 3). All these genes displayed moderate negative correlations, with *p*-values confirming strong statistical associations.

Reactome pathway analysis was performed to explore the biological functions of genes positively correlated with CCND1 in Uterine Corpus Endometrial Carcinoma (UCEC). The analysis revealed several highly enriched pathways, including “Immune System”, “Signal Transduction”, “Gene Expression (Transcription)”, “Extracellular Matrix Organization”, and “Cell Cycle”, indicating the involvement of these genes in key biological processes relevant to cancer progression (Figure 5A). Additionally, Reactome pathway analysis for CCND1 demonstrated that positively correlated genes with CCND1 were primarily associated with “Signaling by Receptor Tyrosine Kinases” (*p* = 1.03 × 10^−10^), “Signal Transduction” (*p* = 1.28 × 10^−8^), “Collagen Degradation” (*p* = 6.59 × 10^−6^), “MAPK Family Signaling Cascades” (*p* = 8.78 × 10^-6^), and “Integrin Cell Surface Interactions” (*p* = 1.89 × 10^−5^) (Figure 5B).

Similarly, Reactome pathway analysis was conducted for genes negatively correlated with CCND1 in UCEC, revealing enrichment in pathways such as “Signal Transduction”, “Gene Expression (Transcription)”, “Metabolism”, “Cell Cycle”, and “Immune System” (Figure 6A). Reactome pathway analysis for CCND1 also demonstrated that negatively correlated genes with CCND1 were primarily associated with “SHC1 Events in ERBB2 Signaling” (*p* = 4.23 × 10^−7^), “Oncogene Induced Senescence” (*p* = 8.96 × 10^−7^), “ERBB2 Activates PTK6 Signaling” (*p* = 1.02 × 10^−6^), “ERBB2 Regulates Cell Motility” (*p* = 1.26 × 10^−6^), and “Nuclear Signaling by ERBB4” (*p* = 1.55 ×^−6^) (Figure 6B).

The protein–protein interaction (PPI) networks for positively correlated genes with CCND1 were constructed using STRING and Cytoscape. The analysis identified 50 nodes forming a network with 470 edges, with strong enrichment (PPI enrichment *p*-value < 1.0 × 10^−16^) and a local clustering coefficient of 0.639. In the PPI network (Figure 7A), genes strongly interacting with CCND1 are shown, forming a well-connected network. Using the Cytoscape plugin cytoHubba, hub genes were ranked based on their degree of interaction. The top hub genes in this network are highlighted in red, indicating their higher connectivity and importance. The analysis of positively correlated genes with CCND1 identified the top 10 hub genes (Figure 7B).

Similarly, the PPI network for CCND1 negatively correlated genes identified 50 nodes, 324 edges, and a local clustering coefficient of 0.652, with strong enrichment (PPI enrichment *p*-value < 1.0 × 10^−16^). The network illustrates key genes interacting with CCND1, forming a tightly connected system (Figure 8A). The analysis of negatively correlated genes with CCND1 identified the top 10 hub genes (Figure 8B).

Using the KEGG PATHWAY Database, a schematic representation of signaling pathways involved in the development and progression of EC was constructed, with a focus on the regulation of CCND1. In Type I endometrioid adenocarcinoma, mutations in PTEN, K-Ras, and β-Catenin lead to the activation of key pathways such as PI3K-Akt, MAPK, and Wnt, which promote CCND1 overexpression through β-Catenin-mediated TCF/LEF transcription and activation of c-Myc [13,14,15,16]. In Type II serous adenocarcinoma, Her2/neu amplification and p53 mutations drive tumor progression through the ErbB and p53 signaling pathways, respectively (Figure 9) [17,18,19].

GO functional enrichment analysis was conducted on genes positively correlated with CCND1, assessing their roles in biological processes (BP), cellular components (CC), and molecular functions (MF) using the DAVID tool. Significant enrichment was observed in BP terms, including GO:0070373 (negative regulation of ERK1 and ERK2 cascade), GO:1902747 (negative regulation of lens fiber cell differentiation), and GO:0030512 (negative regulation of TGF-β receptor signaling pathway) (Figure 10A). Furthermore, enriched CC terms included GO:0062023 (collagen-containing extracellular matrix), GO:0005788 (endoplasmic reticulum lumen), and GO:0005581 (collagen trimer) (Figure 10B). Notable MF terms identified were GO:0048407 (platelet-derived growth factor binding), GO:0005515 (protein binding), and GO:0030020 (extracellular matrix structural constituent conferring tensile strength) (Figure 10C). These results highlight the critical functional roles of genes positively correlated with CCND1 in cellular regulation and structural dynamics.

Similarly, GO functional enrichment analysis was conducted on genes negatively correlated with CCND1, evaluating their roles in biological processes (BP), cellular components (CC), and molecular functions (MF) using the DAVID tool. Key enriched terms in BP included GO:1902510 (regulation of apoptotic DNA fragmentation), GO:0038138 (ERBB4–ERBB4 signaling pathway), and GO:0050821 (protein stabilization) (Figure 11A). Within the CC category, significant terms were GO:0005634 (nucleus), GO:0005737 (cytoplasm), and GO:0005813 (centrosome) (Figure 11B). In the MF category, the most enriched terms included GO:0005515 (protein binding), GO:0048018 (receptor ligand activity), and GO:0016706 (2-oxoglutarate-dependent dioxygenase activity) (Figure 11C). These results emphasize the regulatory and structural roles of genes negatively correlated with CCND1 in cellular processes.

### 2.6. Relationships to Survival

The prognostic significance of CCND1 mRNA expression was evaluated across multiple survival outcomes, including OS, Progression-Free Survival (PFS), Disease-Specific Survival (DSS), and Disease-Free Survival (DFS), using a uniform cut-off value of 12.22 for all analyses. Kaplan–Meier survival analysis demonstrated a statistically significant association between CCND1 expression and OS in the overall endometrial cancer cohort (*p* = 0.029; Figure 12A). In the endometrioid subtype, low CCND1 expression was significantly correlated with poorer survival outcomes compared to high expression (*p* = 0.0004; Figure 12B). Conversely, no statistically significant relationship was observed between CCND1 expression and OS in the non-endometrioid subtype (*p* = 0.71; Figure 12C). Notably, significant differences in survival were observed between the endometrioid and non-endometrioid subtypes, with high CCND1 expression consistently associated with improved survival outcomes across the significant analyses.

Subsequently, we conducted an analysis to evaluate the impact of CCND1 mRNA expression on PFS in patients with endometrial cancer. Kaplan–Meier survival analysis demonstrated a significant association between CCND1 expression and PFS in the overall cohort (*p* = 0.02; Figure 13A) and in the endometrioid subtype (*p* = 0.006; Figure 13B). However, in the non-endometrioid subtype, no statistically significant relationship was observed (*p* = 0.72; Figure 13C). High CCND1 expression was consistently linked to improved PFS outcomes across significant analyses.

Following this, we analyzed the impact of CCND1 mRNA expression on DFS in patients with EC. Kaplan–Meier survival analysis showed a significant association between CCND1 expression and DFS in the overall cohort (*p* = 0.009; Figure 14A). In the endometrioid subtype, no statistically significant association was observed (*p* = 0.19; Figure 14B). Similarly, in the non-endometrioid subtype, the relationship between CCND1 expression and DFS did not reach statistical significance (*p* = 0.06; Figure 14C). Although high CCND1 expression was linked to better outcomes, statistical significance was only observed in the overall cohort.

Building on these findings, we next assessed the impact of CCND1 mRNA expression on DSS in EC patients. Kaplan–Meier survival analysis demonstrated a significant association in the overall cohort (*p* = 0.03; Figure 15A) and in the endometrioid subtype (*p* = 0.0029; Figure 15B). In contrast, no statistically significant relationship was observed in the non-endometrioid subtype (*p* = 0.87; Figure 15C). High CCND1 expression was consistently linked to improved survival outcomes across significant analyses.

## 3. Discussion

In the present study, we aimed to investigate the utility of CCND1 expression as a prognostic marker in EC by analyzing both our own patient cohort and publicly available in silico datasets in relation to clinicopathological features and patient survival. Initially, we evaluated the immunohistochemical expression of CCND1 protein in tissue samples from EC patients to assess its association with selected clinicopathological traits and overall survival. Subsequently, we performed similar analyses using CCND1 mRNA expression data retrieved from public datasets. Lastly, genes coexpressed with CCND1 in EC were identified and subjected to functional annotation to explore their biological roles and potential pathways involved in cancer progression.

In our study, immunohistochemical analysis revealed that CCND1 expression was predominantly localized to the nuclear compartment, consistent with its anticipated localization in malignant cells. In line with our findings, Tsuda et al. observed weak positive nuclear staining of Cyclin D1 in endometrial cancer tissues [20]. Similarly, Quddus et al., Sangwan et al., and Xu et al. reported nuclear localization of Cyclin D1 in endometrial carcinoma tissues, consistently identifying the nucleus as the primary site of its expression in tumor cells across different cohorts and methodologies [9,21,22].

Kaplan–Meier survival analysis from our study indicated that elevated CCND1 protein levels are more likely to be associated with worse survival outcomes. Similarly, Zapiecki et al. demonstrated that higher Cyclin D1 expression was significantly associated with poorer survival outcomes (*p* = 0.042) [23]. In a related study, Berg et al. further confirmed that high CCND1 expression was linked to worse disease-specific survival, emphasizing its prognostic value in aggressive endometrial cancer cases (*p* = 0.008) [24]. However, Khabaz et al. reported that low Cyclin D1 expression was associated with better survival, while negative Cyclin D1 staining corresponded to significantly poorer survival outcomes, with survival distributions differing markedly between staining categories when adjusted for disease recurrence (*p* = 0.011) [11]. Moreover, Liang et al. highlighted the prognostic relevance of Cyclin D1, reporting that patients with negative expression exhibited higher overall survival rates than those with positive expression (*p* < 0.05) [25]. These findings collectively suggest that Cyclin D1 expression levels play a complex and context-dependent prognostic role in endometrial cancer. While high expression is consistently associated with worse survival outcomes, the absence or low expression of Cyclin D1 appears to have variable effects, potentially influenced by differences in tumor biology, molecular subtypes, and methodological approaches across studies.

Notably, our mRNA analysis revealed a survival trend that contrasts with protein expression patterns. Elevated CCND1 protein levels were associated with worse survival outcomes. Conversely, in the mRNA analysis, lower CCND1 expression significantly correlated with poorer survival, specifically in the endometrioid subtype of endometrial cancer (*p* = 0.0004). In contrast, no significant association was observed in the non-endometrioid subtype (*p* = 0.71). Additionally, CCND1 mRNA expression showed borderline statistical relevance for overall survival in the general endometrial cancer cohort (*p* = 0.029), indicating a limited prognostic value across all subtypes. The discrepancy between CCND1 protein and mRNA survival trends in endometrial cancer may result from post-transcriptional regulation, including mRNA stability controlled by microRNAs or RNA-binding proteins [26,27]. Additionally, differences in protein translation efficiency and degradation through the ubiquitin–proteasome system could explain reduced protein levels despite high mRNA expression [28,29,30]. Methodological factors, such as differences in detection techniques (Immunohistochemical (IHC) vs. RNA sequencing), may further contribute to these variations. These findings underscore the importance of integrating multi-omics approaches combining transcriptomic, proteomic, and clinical data to better understand the regulatory dynamics and prognostic implications of CCND1 in endometrial cancer.

Our analysis of CCND1 expression revealed a gradual increase in positivity with histological grade, from 55.56% in Grade 1 tumors to 73.91% in Grade 3 tumors; however, these differences were not statistically significant (*p* = 0.2090). Similarly, Tsuda et al. observed an increase in Cyclin D1 positivity with higher histological grade, from 17.6% in Grade 1 tumors to 38.5% in Grade 2 and 40.0% in Grade 3 tumors. However, consistent with our findings, these differences were not statistically significant [20]. Supporting these observations, Khabaz et al. also found no significant differences in Cyclin D1 expression across tumor grades [11]. In the same context, Sangwan et al. reported no statistically significant differences in Cyclin D1 expression across tumor grades [9]. Likewise, Yildirim et al. reported a statistically significant increase in Cyclin D1 expression with advancing histological grade, observing 61% positivity in Grade 1 tumors, 94% in Grade 2, and 100% in Grade 3 (*p* < 0.001) [31]. These findings collectively suggest that while Cyclin D1 expression may correlate with tumor grade, its significance could depend on cohort-specific factors, such as sample size, methodology, or tumor heterogeneity.

For FIGO stages, our immunohistochemical analysis demonstrated the highest CCND1 positivity in Stage IV tumors (90.9%), with lower levels in Stage I (75.41%) and Stage II (76.67%), though these differences were not statistically significant (*p* = 0.6862). Notably, Khabaz et al. observed a statistically higher frequency of negative Cyclin D1 staining in FIGO stages III and IV, suggesting a potential shift in Cyclin D1 expression patterns with disease progression (*p* = 0.029) [11]. Zapiecki et al. reported a similar trend, where Cyclin D1 expression, quantified as the mean percent positive nuclear area (PPNA), was significantly higher in advanced stages (Stage I: 4.3, Stage IV: 17.5; *p* = 0.013), indicating its potential role in tumor progression [23]. Yildirim et al. observed an increase in positivity with advancing FIGO stages, reporting 60% in Stage I, 93% in Stage II, and 92% in Stage III (*p* < 0.001) [31]. Similarly, Berg et al. found high CCND1 expression predominantly in FIGO stages III and IV, with 57.1% and 75% positivity, respectively, compared to only 19.4% in Stage I (*p* < 0.001) [24]. Our findings, along with results reported in other studies, demonstrate a general trend of increasing CCND1 expression with higher FIGO stages. Most studies observed elevated CCND1 levels in advanced-stage tumors, supporting its potential role in promoting tumor progression through increased proliferative signaling and survival mechanisms. Despite some variability due to methodological differences and limited statistical significance, the overall trend indicates that CCND1 upregulation may be linked to more aggressive tumor behavior in advanced endometrial cancer stages, warranting further investigation in larger, well-defined cohorts.

For lymph node status (pN), our study showed comparable CCND1 positivity between N0 cases (75.47%) and N1 cases (81.82%), with no statistically significant difference (*p* = 0.7823). By comparison, Yildirim et al. observed a statistically significant increase in CCND1 positivity in N1 cases (92%) compared to N0 cases (75%), indicating a strong association with lymph node invasion (*p* < 0.001) [31]. Similarly, Berg et al. demonstrated a statistically significant link between high CCND1 expression and lymph node metastases (*p* < 0.001) [24]. Although our findings did not reach statistical significance, we observed a trend toward increased CCND1 expression in cases with lymph node metastases, consistent with significant associations reported in other studies, suggesting a potential role for CCND1 in promoting tumor progression.

Regarding histological type, high CCND1 expression in our cohort was similar between endometrioid (77.45%) and non-endometrioid cancers (73.08%), with no statistically significant difference observed (*p* = 0.6136). In contrast, Khabaz et al. reported statistically significant differences in Cyclin D1 expression across histological subtypes, with higher expression in endometrioid adenocarcinomas (20.3%) [11]. Zapiecki et al. found that Cyclin D1 expression varied across histological subtypes, being most pronounced in adenosquamous carcinomas, with a mean PPNA of 14.8 (*p* = 0.028), highlighting subtype-specific differences [23]. Atram et al. further distinguished Cyclin D1 expression by Type I and Type II endometrial cancer classifications, reporting that Cyclin D1 expression was predominantly associated with Type I tumors, corresponding to endometrioid adenocarcinomas, particularly those with poor differentiation. In their cohort of 124 EC cases, 87.09% were classified as Type I (endometrioid) and 12.89% as Type II (non-endometrioid), with an overall Cyclin D1 positivity of 53.22% [32]. Yildirim et al., however, reported significantly higher Cyclin D1 expression in endometrioid carcinoma (78%) compared to precursor lesions, such as simple hyperplasia (30%) and endometrial intraepithelial neoplasia (62%) (*p* < 0.001) [31]. While most studies focus on the protein expression of Cyclin D1, Lapke et al. highlighted alterations in CCND1 at the genetic level, particularly changes in copy number variations (CNVs). These alterations were observed less frequently in endometrioid endometrial cancer compared to other subtypes, such as high-grade serous carcinomas, where CCND1 CNVs were significantly more common (*p* = 0.007) [33]. Notably, this aligns with the observation that endometrioid tumors often exhibit less genomic instability compared to serous subtypes [34]. These findings suggest that genetic changes in CCND1 may not always translate directly into alterations in protein-level expression.

Our findings indicated no significant association between Cyclin D1 expression and patient age, as positivity was observed in 75.61% of patients aged ≤60 years and 77.01% of patients aged >60 years (*p* > 0.8613). In contrast, Khabaz et al. reported that high Cyclin D1 expression was more prevalent in younger patients (<40 years old) and decreased with age (*p* = 0.0001), highlighting a potential demographic influence on Cyclin D1 expression in endometrial cancer [11]. These results suggest that the relationship between Cyclin D1 expression and patient age may be context-dependent, potentially influenced by tumor-specific molecular characteristics and individual patient factors.

The pathway enrichment analysis of genes coexpressed with CCND1 suggests its involvement in processes beyond cell cycle regulation, particularly in pathways related to signal transduction, immune modulation, and extracellular matrix remodeling. These findings imply that CCND1 may influence both tumor-intrinsic mechanisms, such as cell survival and proliferation, and tumor-extrinsic processes, including interactions with the microenvironment that facilitate immune evasion and metastatic progression. The association with ERBB signaling pathways among negatively correlated genes points to a potential compensatory mechanism in more aggressive tumor subtypes, where alternative oncogenic pathways might become activated in response to changes in CCND1 expression. Additionally, the enrichment of immune-related pathways raises the possibility that CCND1 plays a role in shaping the tumor immune microenvironment, which could impact immune surveillance and response to therapies. Although CCND1 is primarily associated with cell cycle regulation, our findings indicate that its role may extend to processes beyond proliferation control, particularly in pathways that influence tumor adaptability, immune interactions, and progression in endometrial cancer.

This study has several limitations that should be acknowledged. The analysis was based on both protein and mRNA expression data, which were not derived from the same experimental samples, and the retrospective nature of the datasets may introduce bias. Additionally, while the study identified significant prognostic associations of CCND1 expression in endometrial cancer, the functional mechanisms underlying these observations were not experimentally validated. Moreover, the lack of experimental verification limits the interpretation of the biological role of CCND1 in tumor progression. It is also important to note that missing values within publicly available datasets could reduce the statistical power of some analyses, potentially affecting the representativeness of the findings. Therefore, further studies on larger, independent cohorts with complete and prospectively collected data are warranted to confirm our observations and better understand the molecular mechanisms related to CCND1 in endometrial cancer.

## 4. Materials and Methods

### 4.1. Patients and Tissue Material

This study was conducted using archived tissue material from the Department of Clinical Pathomorphology, Collegium Medicum in Bydgoszcz, Nicolaus Copernicus University in Toruń, with ethical approval from the Institutional Ethics Committee (KB/87/2020). The study group comprised 128 patients histologically diagnosed with EC who underwent abdominal hysterectomy with bilateral salpingo-oophorectomy and lymphadenectomy at the Department of Obstetrics, Gynecological Diseases, and Oncological Gynecology at Dr. Jan Biziel University Hospital No. 2 in Bydgoszcz. The patients ranged in age from 40 to 84 years, with a mean age of 66.2 years and a median age of 66 years. Postoperative histopathological examination confirmed the diagnosis of EC. To account for the heterogeneous nature of the study group and the limited number of cases within specific categories of the International Federation of Gynecology and Obstetrics (FIGO) classification, a simplified classification was applied to reflect trends in disease progression. Key clinical and pathological variables included age (≤60 vs. >60 years), histologic grade (G1, G2, G3), pathological T stage (pT1–pT4), pathological N stage (pN0–pN1), pathological M stage (pM0–pM1), FIGO staging (I–IV), lymphovascular space invasion (LVSI; present vs. absent), and tumor histology (endometrioid vs. non-endometrioid type). The date of 13 April 2022, was set as the cut-off for overall survival (OS) analysis, defined as the time from diagnosis to the last follow-up or death. The median follow-up duration was 106.5 months, during which 58 patients (45.31%) passed away. Postsurgical survival data were available for all patients.

The control group consisted of material obtained from 30 patients who had previously undergone surgical hysterectomy due to uterine fibroids at the same department. The patients’ ages ranged from 45 to 71 years, with a mean age of 60 years. Histopathological examination of the control tissue revealed normal endometrial tissue with no findings associated with proliferation, inflammation, or neoplastic changes.

### 4.2. Immunohistochemistry

Immunohistochemical analysis of CCND1 expression was conducted using tissue macroarrays (TMAs) prepared from tumor-enriched sections of paraffin-embedded blocks and corresponding histologically normal tissues. Four-micrometer-thick sections were generated using a manual rotary microtome (Accu-Cut, Sakura, Torrance, CA, USA) and mounted on adhesion-coated slides (SuperFrost Plus, Menzel-Gläser, Braunschweig, Germany). Sections were dried at 60 °C for 30 min before further processing. Tissue sections were dewaxed using xylene, rehydrated through graded ethanol series, and subjected to antigen retrieval using Ventana high-pH CC1 buffer (Roche Diagnostics/Ventana Medical Systems, Tucson, AZ, USA) for 64 min. The samples were incubated with a ready-to-use rabbit monoclonal anti-CCND1 antibody (SP4-R, VENTANA) for 24 min. Antigen-antibody complexes were visualized using the UltraView Universal DAB Detection Kit (Ventana Medical Systems, Tucson, AZ, USA). Counterstaining was performed with hematoxylin to highlight nuclear structures, followed by dehydration through graded ethanol concentrations (80%, 90%, 96%, 99.8%), clearing in xylene, and coverslipping with Dako mounting medium (Agilent Technologies, Santa Clara, CA, USA). Positive control sections were selected based on antibody datasheets and the Human Protein Atlas (http://www.proteinatlas.org, accessed on 12 September 2021), while negative controls were prepared by omitting the primary antibody to confirm staining specificity. Microscopic evaluation of CCND1 expression was performed using an ECLIPSE E400 microscope (Nikon, Tokyo, Japan), focusing on the intensity and localization of staining. Morphometric analysis was conducted at 20× magnification, selecting three representative areas per sample to ensure reproducibility and precision in detecting CCND1 expression across all evaluated specimens. This standardized protocol provided reliable and consistent results for CCND1 immunohistochemical assessment.

### 4.3. Immunohistochemical Scoring

Immunohistochemical scoring for CCND1 expression was conducted using the Immunoreactive Score (IRS), which combines the immunointensity score (IS) and immunopercentage score (PS) to provide a semiquantitative evaluation of protein expression levels. Tumor sections were reviewed under a light microscope (ECLIPSE E400, Nikon Instruments Europe, Amsterdam, The Netherlands) at 20× magnification by an experienced pathologist blinded to the clinical and pathological data. For each sample, three randomly selected fields were assessed, and the mean IRS value was calculated. The immunointensity score (IS) represented the staining strength and was graded as follows: 0 = no staining, 1 = weak positive, 2 = moderate positive, and 3 = strong positive. The immunopercentage score (PS) reflected the proportion of positively stained cells, categorized as 0 = no positive cells, 1 = 5–24% positive cells, 2 = 25–49% positive cells, 3 = 50–74% positive cells, and 4 = ≥75% positive cells. The final IRS was calculated by multiplying IS and PS, resulting in values ranging from 0 to 12. High expression of CCND1 was defined as IRS ≥ 1, while low expression corresponded to IRS < 1. To dichotomize expression levels, optimal cut-off points were determined using the cutpoint function of the Evaluate Cutpoints application in R (version 3.4.1, Medical University of Lodz, Lodz, Poland) [35]. This approach allowed for the stratification of samples into “high” and “low” expression groups for subsequent correlation analyses with clinicopathological features and OS. This scoring system provided a standardized and reproducible method for evaluating CCND1 expression in tumor tissues.

### 4.4. Database Analysis

Gene expression data were obtained from the UCSC Xena platform as RSEM expected counts (DESeq2 standardized), preprocessed using the STAR aligner and RSEM quantification, and normalized via DESeq2. The Cancer Genome Atlas (TCGA) cohort comprised 174 samples diagnosed with endometrial cancer and 23 samples of non-cancerous endometrial tissues, which served as controls. Among the 174 cancer samples, 105 were classified as Uterine Endometrioid Carcinoma, while 69 were identified as non-endometrioid subtypes, including 13 cases of Mixed Serous and Endometrioid Carcinoma and 56 cases of Serous Endometrial Adenocarcinoma. In the analyses of protein expression using IHC staining, a cut-off point of 1 was applied, with expression levels equal to or above 1 classified as high expression and below 1 classified as low expression. RNA sequencing (RNA-seq) transcriptome data were retrieved through the UCSC Xena Browser (http://xena.ucsc.edu/ (accessed on 5 October 2024)) and subsequently normalized using the DESeq2 normalization method. The mRNA expression levels were stratified into high and low groups based on cut-off points identified using the Evaluate Cutpoints software. Values below 12.22 for CCND1 were classified as low gene expression, whereas values equal to or exceeding the established cut-off point were considered indicative of high expression. Further analysis was conducted to identify the top 50 genes positively correlated with CCND1 UCEC utilizing the cBioPortal web resource (https://www.cbioportal.org/ (accessed on 9 January 2024)) and the TCGA dataset. Pathway enrichment and visualization were performed using the Reactome Pathway Database (https://reactome.org (accessed on 10 January 2024)), while the Kyoto Encyclopedia of Genes and Genomes (KEGG) Pathway Database (https://www.genome.jp/kegg/pathway.html (accessed on 15 November 2024)) was used to explore the pathways associated with the development and progression of endometrial carcinoma. The STRING database (https://string-db.org (accessed on 10 January 2024)) and Cytoscape software (version 3.10.3, Cytoscape Consortium, San Diego, CA, USA) with the cytoHubba plugin facilitated the construction of a PPI network for the top 50 CCND1-coexpressed genes. The analysis was conducted with a medium confidence score of 0.700, incorporating 300 positively correlated genes and 300 negatively correlated genes. To determine the Gene Ontology (GO) categories, including cellular component (CC), biological process (BP), and molecular function (MF) shared by these genes, the Database for Annotation, Visualization and Integrated Discovery (DAVID; https://david.ncifcrf.gov (accessed on 11 January 2024)) was employed.

### 4.5. Statistical Analysis

All statistical analyses were conducted using GraphPad Prism software (version 7.01, GraphPad Software, La Jolla, CA, USA) and SPSS Statistics Data Editor (version 26.0, IBM Corporation, Chicago, IL, USA). The Shapiro–Wilk test was applied to assess the normality of the data. Due to the non-normal distribution of the analyzed variables, group comparisons were performed using the Mann–Whitney U test for continuous variables. Fisher’s exact test and the Chi-square test were employed to analyze associations between categorical clinical parameters and the expression of CCND1, which is classified as a categorical variable. Survival analyses were performed using the Kaplan–Meier method, and comparisons between survival curves were made with the Log–Rank test. To investigate the effect of CCND1 expression on OS, a Cox proportional hazards regression model was used. Univariate analyses were conducted to calculate hazard ratios (HR) with 95% confidence intervals (95% CI). Spearman’s rank correlation coefficient was used to assess the relationships between CCND1 expression and clinical variables, as well as other biomarkers. Correlation strength was classified according to Guilford’s scale, ranging from no correlation (r = 0) to a full correlation (r = 1). Statistical significance was defined as *p* < 0.05. These methods ensured a comprehensive and rigorous analysis of CCND1 expression and its clinical significance.

## 5. Conclusions

This study highlights the multifaceted role of CCND1 in endometrial cancer, suggesting its involvement in pathways beyond cell cycle regulation, including apoptosis, immune response, and tumor microenvironment modulation. Although our analyses identified potential links between CCND1 overexpression and advanced disease stages, the lack of statistical significance in some results reflects the complexity of its prognostic relevance. Notable limitations include the use of independent protein and mRNA datasets, the retrospective nature of the study, and the absence of experimental validation, which may have impacted data consistency and interpretability. Despite these constraints, our findings lay a foundation for future research aimed at the mechanistic exploration of CCND1 and its associated pathways. To fully elucidate the role of CCND1 in endometrial cancer progression and its potential as a prognostic biomarker, further studies integrating functional assays, multi-omics approaches, and prospectively collected cohorts will be essential.

## Figures and Tables

**Figure 1 ijms-26-00890-f001:**
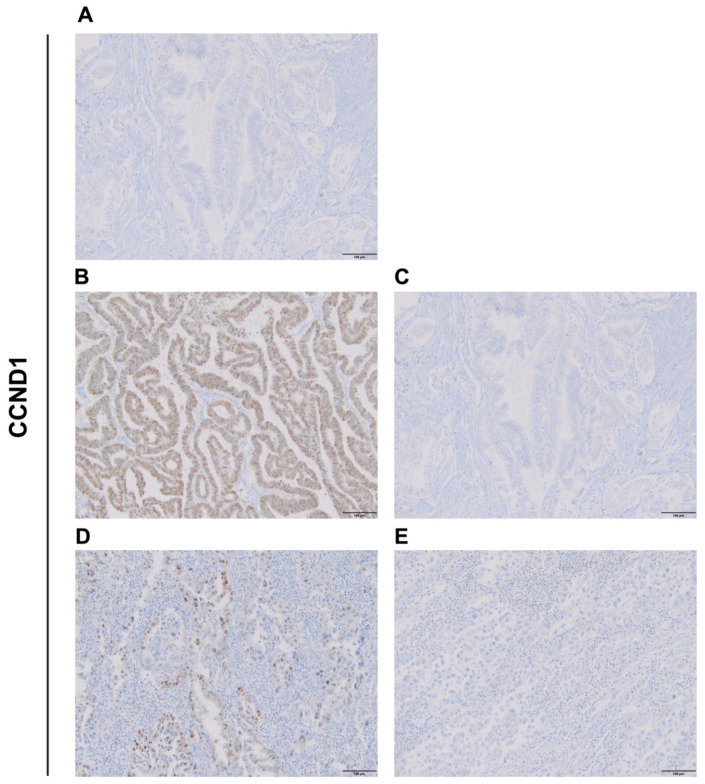
Representative photographs showing immunohistochemical expression of CCND1 in EC. (**A**) Negative control for Cyclin D1, showing no detectable staining in normal endometrial tissue. (**B**) Endometrioid carcinoma with low CCND1 expression. (**C**) Endometrioid carcinoma with strong CCND1 expression. (**D**) Serous carcinoma with high CCND1 expression. (**E**) Serous carcinoma with low CCND1 expression. Original magnification 20×.

**Figure 2 ijms-26-00890-f002:**
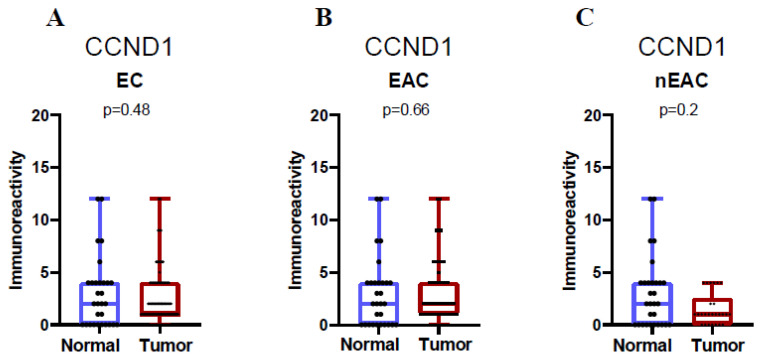
Immunoexpression of CCND1 in EC. The X-axis of the plot represents histologically normal endometrial tissue vs. cancerous endometrial tissue. Data for all endometrial cancers (**A**), endometrioid cancer (**B**), and non-endometrioid cancer (**C**). The Y-axis represents immunoreactivity IRS. The top and bottom of the error bars represent the maximum and minimum values of data, respectively.

**Figure 3 ijms-26-00890-f003:**
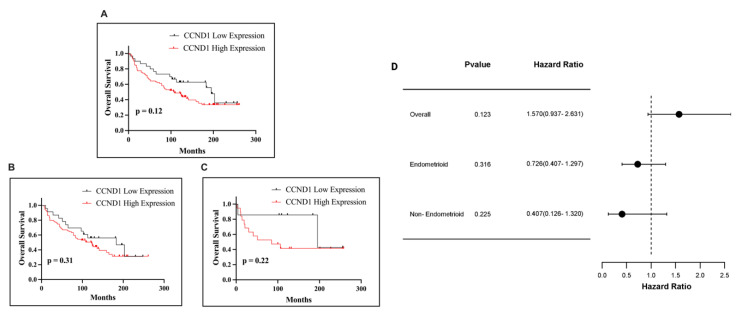
Kaplan–Meier curves presenting the OS of EC patients based on immunohistochemical CCND1 expression for all cases combined (**A**), endometrioid subtype (**B**), and non-endometrioid subtype (**C**). Forest plot of hazard ratios (HR) with 95% confidence intervals (CI) for CCND1 expression in EC (**D**).Green dots represent statistically significant results (*p* < 0.05) with a HR > 1, while black dots indicate non-significant results (*p* > 0.05) regardless of whether HR > 1 or HR < 1. The same annotation scheme applies to all Kaplan-Meier survival curves.

**Figure 4 ijms-26-00890-f004:**
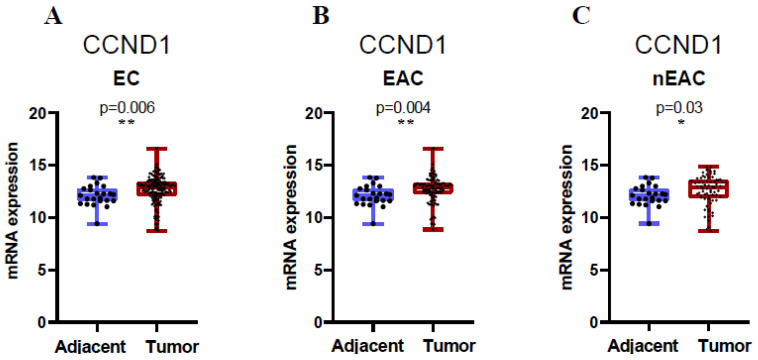
mRNA expression of CCND1 in endometrial cancer subtypes. mRNA expression data were retrieved from the Cancer Genome Atlas (TCGA). The X-axis of the plot represents adjacent normal tissue vs. cancer tissue, and the Y-axis represents normalized mRNA expression levels. All ECs (**A**), endometrioid adenocarcinoma (EAC) (**B**), and non-endometrioid adenocarcinoma (nEAC) (**C**). The top and bottom of the error bars represent the maximum and minimum values of the data, respectively. Asterisks indicate statistical significance (** *p* < 0.01, * *p* < 0.05, Mann–Whitney test).

**Figure 5 ijms-26-00890-f005:**
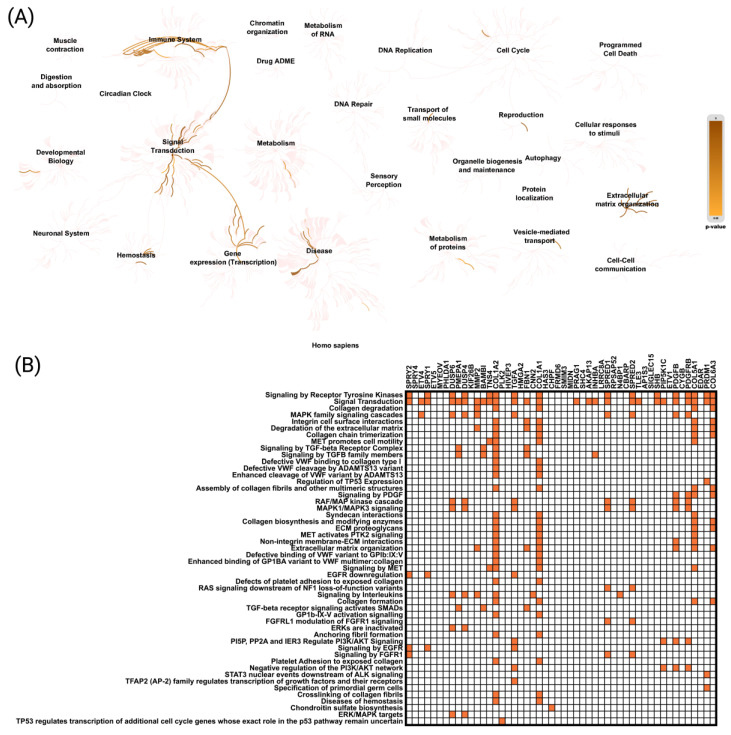
Functional enrichment analysis based on the TCGA dataset, cBioPortal web tool, and Reactome database. The analysis highlights the top 50 genes positively correlated with CCND1 and their involvement in key Reactome pathways associated with CCND1 expression (**A**,**B**).

**Figure 6 ijms-26-00890-f006:**
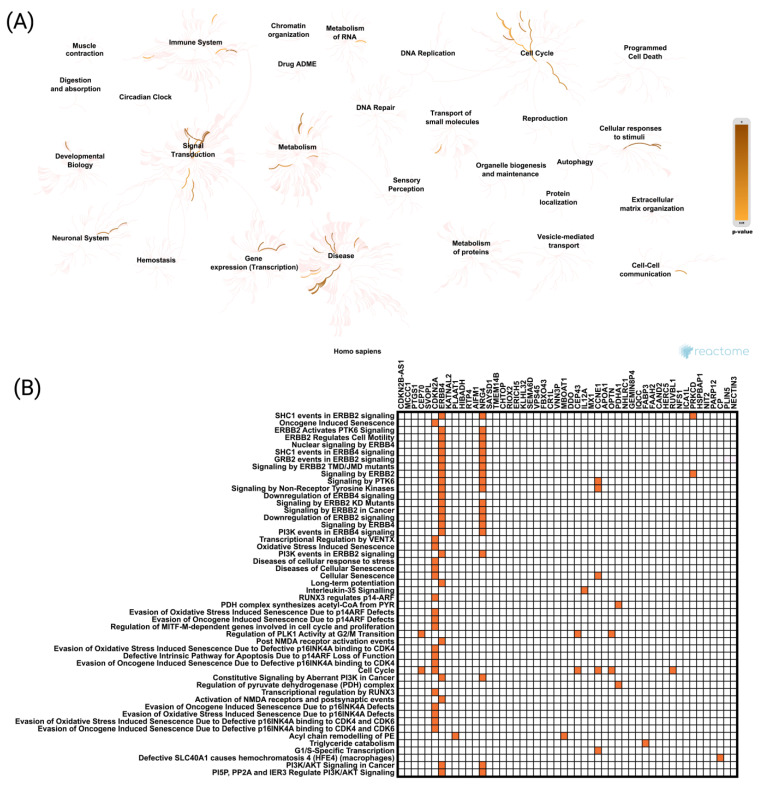
Functional enrichment analysis based on the TCGA dataset, cBioPortal web tool, and Reactome database. The analysis highlights the top 50 genes negatively correlated with CCND1 and their involvement in key Reactome pathways associated with CCND1 expression (**A**,**B**).

**Figure 7 ijms-26-00890-f007:**
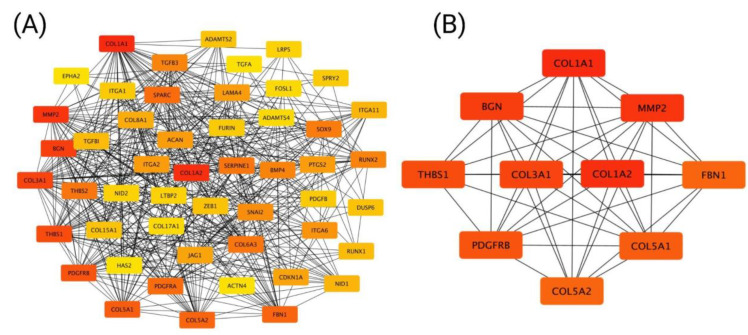
Protein–protein interaction network for genes positively correlated with CCND1. Network of the top 50 genes (**A**) and the top 10 hub genes (**B**).

**Figure 8 ijms-26-00890-f008:**
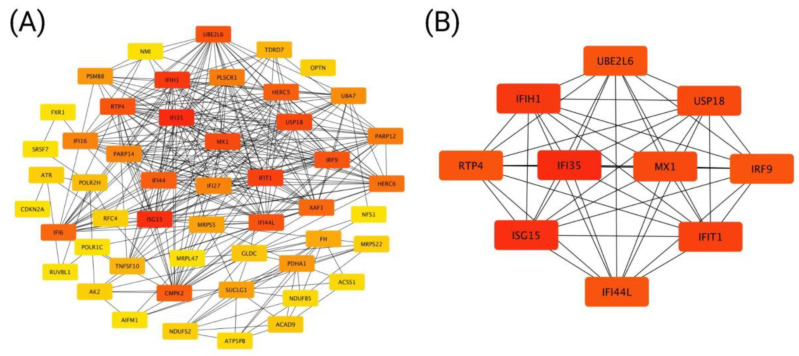
Protein–protein interaction network for genes negatively correlated with CCND1. Network of the top 50 genes (**A**) and the top 10 hub genes (**B**).

**Figure 9 ijms-26-00890-f009:**
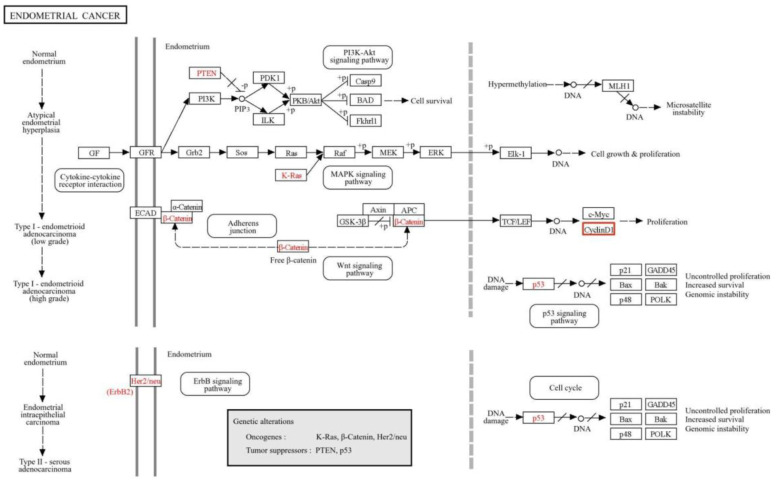
The Figure illustrates simplified pathways involved in the development and progression of EC based on KEGG BRITE data (BRITE accession ID: K04503, pathway map ID: map05213). It outlines the molecular alterations in signaling pathways and genetic changes leading to Type I (endometrioid adenocarcinoma) and Type II (serous adenocarcinoma) EC, with a focus on Cyclin D1 overexpression. In Type I EC, Cyclin D1 is upregulated through β–Catenin-mediated TCF/LEF transcription in the Wnt signaling pathway and is influenced by the MAPK pathway. In Type II EC, Cyclin D1 overexpression is associated with alterations in the p53 and ErbB signaling pathways.

**Figure 10 ijms-26-00890-f010:**
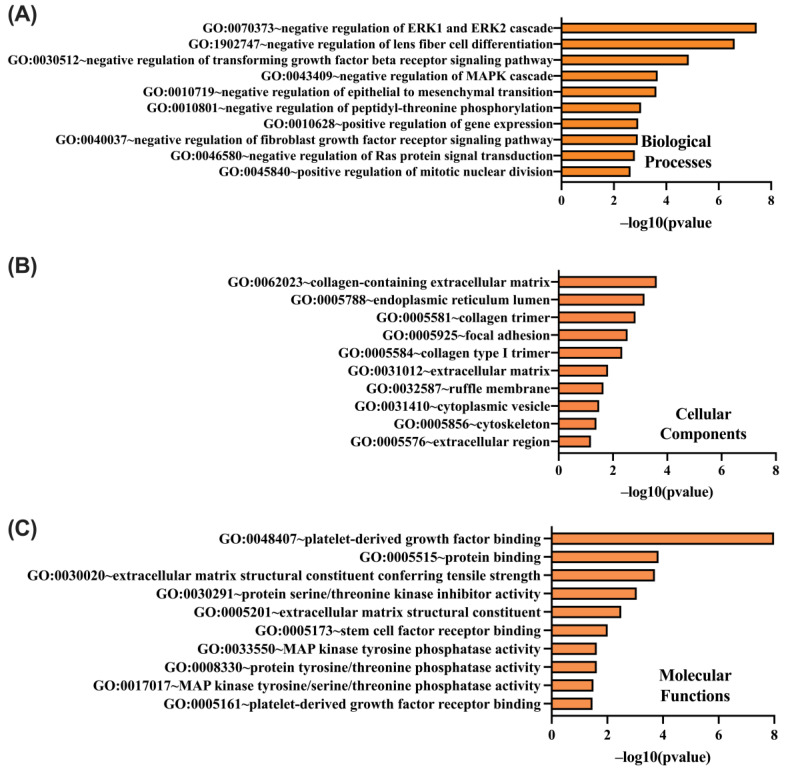
DAVID functional Gene Ontology (GO) analysis of positively correlated genes with CCND1, categorized into BP (**A**), CC (**B**), and MF (**C**). The top 10 GO terms are shown for each category, with *p*-values calculated and ranked based on –log10(*p*-value).

**Figure 11 ijms-26-00890-f011:**
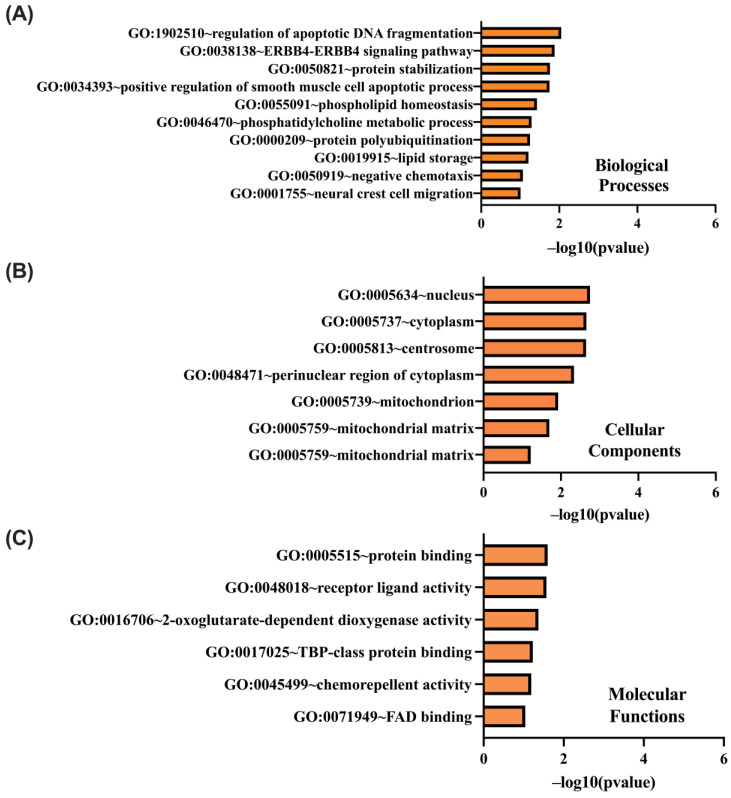
DAVID functional Gene Ontology (GO) analysis of negatively correlated genes with CCND1, categorized into BP (**A**), CC (**B**), and MF (**C**). GO terms are ranked based on –log10(*p*-value), highlighting significant enrichment in each category.

**Figure 12 ijms-26-00890-f012:**
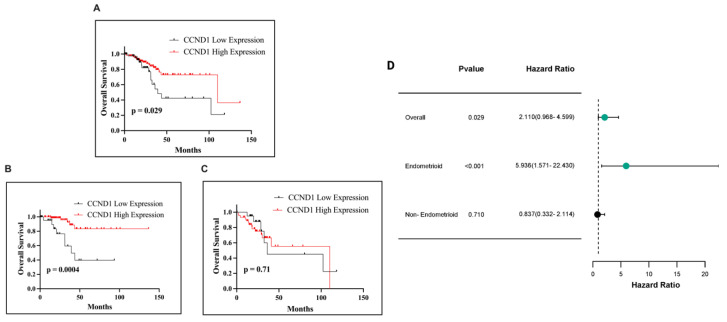
Kaplan–Meier survival curves illustrating OS in EC patients based on CCND1 expression. OS in all EC patients (**A**), in patients with the endometrioid subtype (**B**), and in patients with the non-endometrioid subtype (**C**). Forest plot of HR with 95% CI for CCND1 expression in EC (**D**).

**Figure 13 ijms-26-00890-f013:**
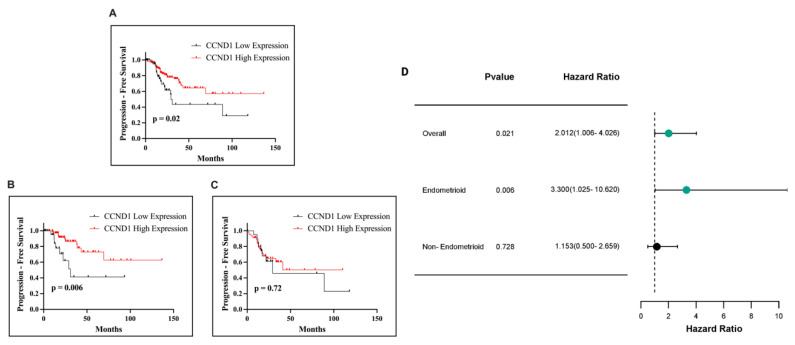
Kaplan–Meier survival curves illustrating PFS in EC patients based on CCND1 expression. PFS in all EC patients (**A**), in patients with the endometrioid subtype (**B**), and in patients with the non-endometrioid subtype (**C**). Forest plot of HR with 95% CI for CCND1 expression in EC (**D**).

**Figure 14 ijms-26-00890-f014:**
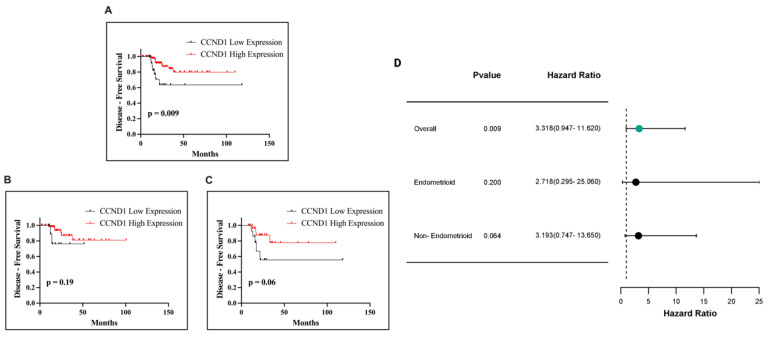
Kaplan–Meier survival curves illustrating DFS in EC patients based on CCND1 expression. DFS in all EC patients (**A**), in patients with the endometrioid subtype (**B**), and in patients with the non-endometrioid subtype (**C**). Forest plot of HR with 95% CI for CCND1 expression in EC (**D**).

**Figure 15 ijms-26-00890-f015:**
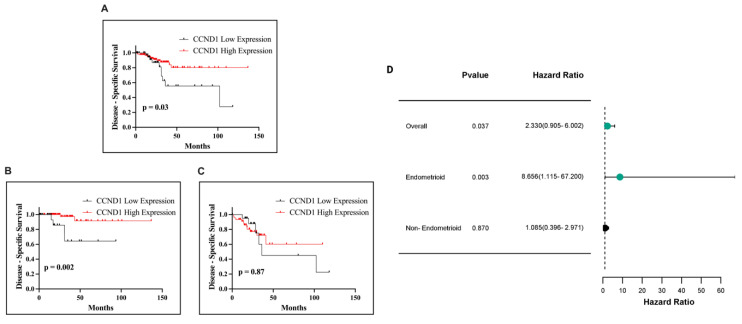
Kaplan–Meier survival curves illustrating DSS in EC patients based on CCND1 expression. DSS in all EC patients (**A**), in patients with the endometrioid subtype (**B**), and in patients with the non-endometrioid subtype (**C**). Forest plot of HR with 95% CI for CCND1 expression in EC (**D**).

**Table 1 ijms-26-00890-t001:** Characteristics of the study population by CCND1 immunoexpression groups. The upward arrow (↑) represents high expression, while the downward arrow (↓) denotes low expression of CCND1.

Cyclin D1				
Variables	Number (%) n = 128	↑n = 98	↓n = 30	*p*-Value
Age				
≤60	41 (32.03)	31 (75.61)	10 (24.39)	0.8613
>60	87 (67.97)	67 (77.01)	20 (22.99)	
Histological grade				
G1	9 (7.03)	5 (55.56)	4 (44.45)	0.2090
G2	73 (57.03)	59 (80.82)	14 (46.67)	
G3	46 (35.94)	34 (73.91)	12 (26.08)	
pT status				
T1	70 (54.69)	53 (75.71)	17 (24.29)	0.7741
T2	36 (28.12)	29 (80.56)	7 (19.44)	
T3	17 (13.28)	13 (76.47)	4 (23.53)	
T4	5 (3.91)	3 (60)	2 (40)	
pN status				
N0	106 (82.81)	80 (75.47)	26 (24.54)	0.7823
N1	22 (17.19)	18 (81.82)	4 (18.18)	
pM status				
M0	116 (90.63)	88 (75.86)	28 (25.14)	0.7307
M1	12 (9.37)	10 (83.33)	2 (16.67)	
FIGO				
I	61 (47.66)	46 (75.41)	15 (24.59)	0.6862
II	30 (23.44)	23 (76.67)	7 (23.33)	
III	26 (20.31)	19 (73.08)	7 (26.92)	
IV	11 (8.59)	10 (90.9)	1 (9.09)	
LVSI				
N	103 (80.47)	77 (74.76)	26 (25.24)	0.4347
T	25 (19.53)	21 (84.00)	4 (16.00)	
Histological type				
Endometrioid cancer	102 (79.69)	79 (77.45)	23 (22.55)	0.6136
Non-endometrioid cancer	26 (20.31)	19 (73.08)	7 (26.92)	

**Table 2 ijms-26-00890-t002:** Genes positively correlated with CCND1.

CCND1 (+) Correlated Gene	Cytoband	Spearman’s Correlation	*p*-Value	CCND1 (+) Correlated Gene	Cytoband	Spearman’s Correlation	*p*-Value
**SPRY2**	13q31.1	0.519	1.08 × 10^–37^	**SMIM3**	5q33.1	0.332	4.72 × 10^–15^
**SPRY4**	5q31.3	0.465	1.27 × 10^–29^	**MIDN**	19p13.3	0.332	4.74 × 10^−15^
**ETV4**	17q21.31	0.440	2.59 × 10^–26^	**PRAG1**	8p23.1	0.332	4.91 × 10^−15^
**SPRY1**	4q28.1	0.420	6.26 × 10^–24^	**SHC4**	15q21.1	0.332	5.02 × 10^−15^
**ETV5**	3q27.2	0.418	1.00 × 10^–23^	**AKAP13**	15q25.3	0.332	5.19 × 10^−15^
**MYEOV**	11q13.3	0.418	1.17 × 10^–23^	**INHBA**	7p14.1	0.330	7.93 × 10^−15^
**PHLDA1**	12q21.2	0.417	1.33 × 10^–23^	**LRRC8A**	9q34.11	0.329	8.46 × 10^−15^
**DUSP6**	12q21.33	0.403	5.27 × 10^–22^	**SPRED1**	15q14	0.328	1.03 × 10^−14^
**PMEPA1**	20q13.31	0.400	1.24 × 10^–21^	**RPSAP52**	12q14.3	0.328	1.08 × 10^−14^
**DUSP4**	8p12	0.388	1.99 × 10^–20^	**N4BP1**	16q12.1	0.327	1.44 × 10^−14^
**KIF26B**	1q44	0.366	4.16 × 10^–18^	**CBARP**	19p13.3	0.324	2.41 × 10^−14^
**MMP2**	16q12.2	0.360	1.44 × 10^–17^	**SPRED2**	2p14	0.323	2.91 × 10^−14^
**BAMBI**	10p12.1	0.359	1.66 × 10^–17^	**TLE3**	15q23	0.321	4.56 × 10^−14^
**TNS4**	17q21.2	0.355	4.53 × 10^–17^	**AP1S3**	2q36.1	0.319	6.47 × 10^−14^
**COL1A2**	7q21.3	0.354	5.85 × 10^–17^	**SIGLEC15**	18q21.1	0.319	6.85 × 10^−14^
**PLK2**	5q11.2	0.353	7.21 × 10^–17^	**SHB**	9p13.1	0.317	8.60 × 10^−14^
**HIVEP3**	1p34.2	0.351	9.31 × 10^–17^	**PIP5K1C**	19p13.3	0.317	9.81 × 10^−14^
**TGFA**	2p13.3	0.349	1.46 × 10^–16^	**ETV1**	7p21.2	0.317	9.94 × 10^−14^
**HMGA2**	12q14.3	0.340	1.04 × 10^–15^	**PDGFB**	22q13.1	0.316	1.01 × 10^−13^
**FBN1**	15q21.1	0.339	1.13 × 10^–15^	**CYGB**	17q25.1	0.316	1.13 × 10^−13^
**CNN2**	19p13.3	0.339	1.33 × 10^–15^	**PDGFRB**	5q32	0.315	1.32 × 10^−13^
**COL1A1**	17q21.33	0.338	1.50 × 10^–15^	**COL5A1**	9q34.3	0.315	1.38 × 10^−13^
**HAS3**	16q22.1	0.337	1.68 × 10^–15^	**EDAR**	2q13	0.313	1.78 × 10^−13^
**CHPF**	2q35	0.336	2.04 × 10^–15^	**PRDM1**	6q21	0.313	1.97 × 10^−13^
**FRMD6**	14q22.1	0.335	2.73 × 10^–15^	**COL6A3**	2q37.3	0.313	2.09 × 10^−13^

**Table 3 ijms-26-00890-t003:** Genes negatively correlated with CCND1.

CCND1 (−) Correlated Gene	Cytoband	Spearman’s Correlation	*p*-Value	CCND1 (−) Correlated Gene	Cytoband	Spearman’s Correlation	*p*-Value
**CDKN2B-AS1**	9p21.3	–0.400	1.20 × 10^−21^	**DDO**	6q21	−0.289	1.36 × 10^−11^
**MCCC1**	3q27.1	−0.354	5.83 × 10^−17^	**CEP43**	6q27	−0.289	1.38 × 10^−11^
**PTGS1**	9q33.2	−0.352	8.03 × 10^−17^	**IL12A**	3q25.33	−0.288	1.49 × 10^−11^
**CEP70**	3q22.3	−0.346	2.67 × 10^−16^	**MX1**	21q22.3	−0.284	3.34 × 10^−11^
**SVOPL**	7q34	−0.346	3.03 × 10^−16^	**CCNE1**	19q12	−0.282	4.14 × 10^−11^
**CDKN2A**	9p21.3	−0.331	5.79 × 10^−15^	**APOA1**	11q23.3	−0.281	4.91 × 10^−11^
**ERBB4**	2q34	−0.328	1.15 × 10^−14^	**OPTN**	10p13	−0.281	5.15 × 10^−11^
**KATNAL2**	18q21.1	−0.325	1.93 × 10^−14^	**PDHA1**	Xp22.12	−0.280	6.09 × 10^−11^
**PLAAT1**	3q29	−0.315	1.37 × 10^−13^	**NHLRC1**	6p22.3	−0.279	6.69 × 10^−11^
**HIBADH**	7p15.2	−0.314	1.63 × 10^−13^	**GEMIN8P4**	1p22.2	−0.277	9.21 × 10^−11^
**RTP4**	3q27.3	−0.311	2.90 × 10^−13^	**IQCC**	1p35.2	−0.276	1.08 × 10^−10^
**AIFM1**	Xq26.1	−0.309	4.09 × 10^−13^	**FABP3**	1p35.2	−0.276	1.14 × 10^−10^
**NRG4**	15q24.2	−0.309	4.26 × 10^−13^	**FAAH2**	Xp11.21	−0.276	1.23 × 10^−10^
**SAYSD1**	6p21.2	−0.302	1.42 × 10^−12^	**CAND2**	3p25.2	−0.275	1.32 × 10^−10^
**TMEM14B**	6p24.2	−0.300	2.18 × 10^−12^	**HERC5**	4q22.1	−0.275	1.42 × 10^−10^
**CHTOP**	1q21.3	−0.299	2.22 × 10^−12^	**RUVBL1**	3q21.3	−0.274	1.52 × 10^−10^
**RIOX2**	3q11.2	−0.297	3.58 × 10^−12^	**NFS1**	20q11.22	−0.274	1.68 × 10^−10^
**ERICH5**	8q22.2	−0.295	4.50 × 10^−12^	**ICA1L**	2q33.2	−0.273	1.75 × 10^−10^
**KLHL32**	6q16.1	−0.293	6.72 × 10^−12^	**PRKCD**	3p21.1	−0.272	2.19 × 10^−10^
**SEMA6D**	15q21.1	−0.291	9.11 × 10^−12^	**HSPBAP1**	3q21.1	−0.272	2.33 × 10^−10^
**VPS45**	1q21.2	−0.291	9.15 × 10^−12^	**NIT2**	3q12.2	−0.269	3.52 × 10^−10^
**FBXO43**	8q22.2	−0.291	9.59 × 10^−12^	**PARP12**	7q34	−0.269	3.69 × 10^−10^
**CR1L**	1q32.2	−0.291	1.01 × 10^−11^	**CP**	3q24-q25.1	−0.269	3.72 × 10^–10^
**VNN3P**	6q23.2	−0.291	1.03 × 10^−11^	**PLIN5**	19p13.3	−0.269	3.73 × 10^–10^
**MBOAT1**	6p22.3	−0.289	1.26 × 10^–11^	**NECTIN3**	3q13.13	−0.268	4.21 × 10^–10^

## Data Availability

Publicly available datasets were analyzed in this study. These data can be found here: https://www.cbioportal.org/results/coexpression?cancer_study_list=ucec_tcga_pan_can_atlas_2018&tab_index=tab_visualize&case_set_id=ucec_tcga_pan_can_atlas_2018_all&Action=Submit&gene_list=CCND1&pathways_source=PathwayMapper&comparison_subtab=mrna (accessed on 9 January 2024); https://xenabrowser.net (accessed on 5 October 2024). Our own data presented in this study are available on request from the corresponding author. The data are not publicly available due to ethical restrictions.

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
