# Peer review of "Assessment of Cyclin D1 Expression: Prognostic Value and Functional Insights in Endometrial Cancer: In Silico Study"

_ijms, 2025, doi:10.3390/ijms26030890_

Round 1
Reviewer 1 Report
Comments and Suggestions for Authors
Dear Authors, an interesting original study on the prognostic value of cyclin D1 in endometrial cancer. Before I can provide my definitive decision, it is imperative to address some concerns. Thus, please answer or consider the following:
(1) Materials and Methods, line 93: explain “FIGO” on the first use (International Federation of Gynecology and Obstetrics [Fédération Internationale de Gynécologie et d’Obstétrique]).
(2) Materials and Methods, line 99: were you able to acquire/calculate other endpoints than Overall Survival (OS) such as DFS, PFS, MFS, RFS, DFI, PFI, etc. (at least the first two)? For part utilizing TCGA, it can be easily acquired but I am unsure about your study group of 128 patients. Events due to disease recurrence/progression occur earlier than death from the disease. Moreover, OS is predominantly indifferent between your stratified groups but there are some trends towards statistical significance. The inclusion of these endpoints would be beneficial for your study.
(3) Materials and Methods, line 114: a space mark missing before “Tissue”.
(4) Materials and Methods, section 2.4: if you acquired transcriptome data from Xena to perform normalization via DESeq2, I presume you used counts, but I did not find this information. When detailing this step, please add if counts were calculated using HTSeq or STAR protocol – it is important for the TCGA cohort.
(5) Materials and Methods, section 2.4: if you decided to identify the top 50 CCND1-correlated genes via UALCAN, did you double-check if they are indeed differentially expressed between groups stratified using a cut-off value of 12.22? The relevance of UALCAN findings is questionable most of the time, especially since it does not provide p-values or negatively correlated genes, as well as can be easily verified as not so pronounced – similar to what you did with verification of correlation values for ETV4 and PTGS2 (it is expected to have different results based on Pearson vs Spearman correlation, but still UALCAN is limited). The use of, e.g., cBioPortal instead of UALCAN is suggested to acquire both positively and negatively correlated genes alongside statistical significance, which would help you in reassessing gene ontology via Reactome and networks via STRING/Cytoscape/cytoHubba. Especially since UALCAN findings did not complement the PPI network in Figure 6 (there are no ETV4 and PTGS2 in the top 10 hub genes).
(6) Materials and Methods, line 175: what interaction score was used for STRING? If it is not mentioned, I presume the default one (medium confidence: 0.400), which is quite low but still acceptable. But it must be specified. Or maybe it is the one in line 318, i.e., “a local clustering coefficient of 0.424”? But its annotation suggests otherwise, and parameters should be included in Methods, not Results.
(7) Materials and Methods, section 2.5: I see a mention of Cox regression models and hazard ratios, but they are not visible in the Results. Please include hazard ratios and forest plots whenever you visualize Kaplan-Meier curves.
(8) Results, section 3.1, Figure 1: I cannot understand if representative photographs showing immunohistochemical expression of CCND1 are based on your own patients or Human Protein Atlas (HPA)? If HPA, a mention of patient IDs and antibody quality scores must be included. If from your own patients, then additional comparison using HPA data is advised. Independent on the case, include a scale bar in your images.
(9) Results, Figures 2 and 4: are there other non-tumor data available than adjacent tissue? Moreover, please add some empty space between the names of groups and include the sample size for each group. Furthermore, consider enriching your box plots with jittered data points. I also believe that the Adjacent group is always the same, which makes me wonder if you should combine all data from Figure 2 and separately from Figure 4 into one plot and do a pairwise analysis (this may not help in data from Figure 2, but I believe it might work for Figure 4). It will give you a chance to compare endometrial cancer subtypes between each other and not each subtype separately versus adjacent tissue.
(10) Results, section 3.1, lines 232-238: I think some sentences are repeated at the beginning of section 3.2. Please double-check and combine, if necessary, to avoid repetitions.
(11) Results, section 3.3, line 257: I might have missed it but there was no mention of a cut-off value of 1 in methodology. Rather than that, there was only about 12.22.
(12) Results, section 3.4, lines 276-277: I understand that GTEx was used as a source of normal endometrial tissue, but TCGA also has some normal tissue specimens, which would increase the sample size. These data can be acquired via GDC or TCGA-Assembler. Moreover, were data normalized appropriately given that two separate repositories were used?
(13) Results, Figure 4: GTEx is not mentioned, so what was the source of adjacent tissues? Are they not considered normal specimens in this case (while GTEx was a source of such data)?
(14) The entire manuscript, multiple locations: I noticed that you refer to some Figures using only subfigure symbols, and the main figure number is missing. Please put full annotations, for example, “Figure 5A” instead of “Figure A”. Double-check all images.
(15) Figure 5 title: I think that Reactome rather than UALCAN deserves to be mentioned in the main title because these images are from Reactome. Of course UALCAN served as the source of the top 50 correlated genes but this can be explained right after the title, in the description.
(16) Results, lines 315-316: in order to consider genes as “UDEGs”, you must perform a proper differential expression analysis with a threshold for p-values/FDRs and fold-changes, but I believe it is missing in the paper (unless it was a part of DESeq2 utilization – but still was not described). In order to leave “UDEGs” as is, additional details must be included and appropriate groups must be compared (I suggest comparing groups stratified by cut-offs from EvaluateCutpoints). Alternatively, please rename this group of genes or treat them as in earlier sections – as positively correlated to CCND1.
(17) Results, lines 334-344: in addition to referencing the KEGG database, add citations from the literature when describing subtypes of endometrial cancer. This should allow you to reach >30 references in the manuscript.
(18) Results, Figure 7: Please add a unique BRITE accession ID for the KEGG graph in the figure’s description. Moreover, if you intend to perform differential expression analysis (as I discussed in comment no. 16), it would be much more informative to compare specific groups via PathView, which is able to enrich KEGG graphs with fold-changes from differential expression analysis.
(19) Discussion, first sentence: I think the acquisition of your own patients is completely omitted, while it is an important advantage of your approach.
(20) Please double-check if the scheme of references is according to the journal’s requirements. I thought MDPI required in-text citations as numbers in square brackets.
(21) Discussion, lines 411-414: change “were associated with worse survival outcomes” to “are more likely to be associated with worse survival outcomes” because most of your results from survival analysis are statistically insignificant. Moreover, delete the part “reinforcing its prognostic significance” because the p-value of 0.042 is borderline significant.
(22) Sometimes you refer to imprecise p-values (e.g., p < 0.05, p = 0.000, p > 0.9999) when discussing research by other scientific teams. If possible, please provide specific values and provide the first number that is not zero.
(23) Add a full stop in lines 444 and 515.
(24) Discussion, line 441: The “IHC” abbreviation must be introduced earlier if you want to use it here.
(25) Discussion, lines 466 and 497: explain “PPNA”.
(26) Discussion, lines 526-527: if immune responses are involved then I think an additional analysis based on, e.g., TIMER 2.0 could be included to verify this assumption.
(27) Please add limitations of your study.
(28) Sections after Conclusions such as Author Contributions, Funding, Institutional Review Board Statement, Informed Consent Statement, Conflicts of Interest, etc are missing.
Author Response
Comment 1: Materials and Methods, line 93: explain “FIGO” on the first use (International Federation of Gynecology and Obstetrics [Fédération Internationale de Gynécologie et d’Obstétrique]).
Response 1: The abbreviation "FIGO" has been explained on its first use in the manuscript to ensure clarity for the readers.
Comment 2: Materials and Methods, line 99: were you able to acquire/calculate other endpoints than Overall Survival (OS) such as DFS, PFS, MFS, RFS, DFI, PFI, etc. (at least the first two)? For part utilizing TCGA, it can be easily acquired but I am unsure about your study group of 128 patients. Events due to disease recurrence/progression occur earlier than death from the disease. Moreover, OS is predominantly indifferent between your stratified groups but there are some trends towards statistical significance. The inclusion of these endpoints would be beneficial for your study.
Response 2: We have included Disease-Free Survival (DFS), Progression-Free Survival (PFS), and Disease-Specific Survival (DSS) analyses using the TCGA dataset, and these results have been incorporated into the manuscript. Unfortunately, data for other endpoints such as MFS, RFS, DFI, and PFI were not available in either the TCGA dataset or our independent cohort of 128 patients.
Comment 3: Materials and Methods, line 114: a space mark missing before “Tissue”.
Response 3: The missing space before "Tissue" has been corrected.
Comment 4: Materials and Methods, section 2.4: if you acquired transcriptome data from Xena to perform normalization via DESeq2, I presume you used counts, but I did not find this information. When detailing this step, please add if counts were calculated using HTSeq or STAR protocol – it is important for the TCGA cohort.
Response 4: The RNA-Seq transcriptome data used in this study were obtained from the UCSC Xena platform, specifically from the TCGA cohort. The selected dataset corresponds to RSEM expected counts (DESeq2 standardized), as indicated by the UCSC Xena platform. The TCGA consortium processes RNA-seq data using the STAR aligner for read alignment followed by RSEM (RNA-Seq by Expectation-Maximization) for gene expression quantification. Therefore, the counts provided by the Xena platform are already preprocessed using this pipeline, and HTSeq was not used at any stage of our analysis. This clarification has been added to the Materials and Methods section to ensure transparency in the data processing workflow.
Comment 5: Materials and Methods, section 2.4: if you decided to identify the top 50 CCND1-correlated genes via UALCAN, did you double-check if they are indeed differentially expressed between groups stratified using a cut-off value of 12.22? The relevance of UALCAN findings is questionable most of the time, especially since it does not provide p-values or negatively correlated genes, as well as can be easily verified as not so pronounced – similar to what you did with verification of correlation values for ETV4 and PTGS2 (it is expected to have different results based on Pearson vs Spearman correlation, but still UALCAN is limited). The use of, e.g., cBioPortal instead of UALCAN is suggested to acquire both positively and negatively correlated genes alongside statistical significance, which would help you in reassessing gene ontology via Reactome and networks via STRING/Cytoscape/cytoHubba. Especially since UALCAN findings did not complement the PPI network in Figure 6 (there are no ETV4 and PTGS2 in the top 10 hub genes).
Response 5: We agree with the reviewer that UALCAN findings may lack reliability due to the absence of p-values and negatively correlated genes. Therefore, we have used cBioPortal for all correlation analyses, as it provides both positively and negatively correlated genes along with statistical significance (p-values). Analyses were conducted for both positively and negatively correlated genes, and the results have been reassessed accordingly. Additionally, we used these updated gene lists for Reactome pathway analysis and PPI network construction via STRING and Cytoscape/cytoHubba to ensure the robustness of our findings.
Comment 6: Materials and Methods, line 175: what interaction score was used for STRING? If it is not mentioned, I presume the default one (medium confidence: 0.400), which is quite low but still acceptable. But it must be specified. Or maybe it is the one in line 318, i.e., “a local clustering coefficient of 0.424”? But its annotation suggests otherwise, and parameters should be included in Methods, not Results.
Response 6: The interaction score used for the STRING analysis was 0.700 (high confidence), which has now been specified in the Materials and Methods section. The local clustering coefficient value of 0.424, along with the number of nodes, edges, and the PPI enrichment p-value, was provided in the Results section as part of the network characterization. These parameters have been clarified in the manuscript to avoid confusion.
Comment 7: Materials and Methods, section 2.5: I see a mention of Cox regression models and hazard ratios, but they are not visible in the Results. Please include hazard ratios and forest plots whenever you visualize Kaplan-Meier curves.
Response 7: The Cox regression models and corresponding hazard ratios, along with their 95% confidence intervals and p-values, have been added to all relevant Kaplan-Meier curve analyses in the Results section. Additionally, forest plots summarizing these hazard ratios have been included for each analysis to improve clarity and visualization.
Comment 8: Results, section 3.1, Figure 1: I cannot understand if representative photographs showing immunohistochemical expression of CCND1 are based on your own patients or Human Protein Atlas (HPA)? If HPA, a mention of patient IDs and antibody quality scores must be included. If from your own patients, then additional comparison using HPA data is advised. Independent on the case, include a scale bar in your images.
Response 8: The representative photographs showing immunohistochemical expression of CCND1 are based on our own patient cohort. Following the reviewer's suggestion, scale bars have been added to all images, and an additional image representing high CCND1 expression in serous carcinoma has been included. We explored the Human Protein Atlas (HPA) for comparison; however, we were unable to find representative images of cancerous tissues with CCND1 expression in either endometrioid or non-endometrioid subtypes, particularly in serous or mixed carcinoma, which were prevalent in our cohort. Therefore, we did not include HPA images in the manuscript. Nevertheless, we recognize the value of such a comparison and will make efforts to include HPA data in future publications if relevant images become available.
Comment 9: Results, Figures 2 and 4: are there other non-tumor data available than adjacent tissue? Moreover, please add some empty space between the names of groups and include the sample size for each group. Furthermore, consider enriching your box plots with jittered data points. I also believe that the Adjacent group is always the same, which makes me wonder if you should combine all data from Figure 2 and separately from Figure 4 into one plot and do a pairwise analysis (this may not help in data from Figure 2, but I believe it might work for Figure 4). It will give you a chance to compare endometrial cancer subtypes between each other and not each subtype separately versus adjacent tissue.
Response 9: Thank you for your valuable comments regarding the figures. The control data in the immunohistochemistry results refer to normal tissues; therefore, we have adjusted the labeling on the graphs to clarify this. Additionally, we have added some empty space between the names of the groups to improve readability. To avoid overcrowding the figures, we have decided to keep the sample sizes presented in the text, specifically in Section 2.4 and Table 1. Moreover, following your suggestion, we have enriched all box plots with jittered data points to provide a clearer representation of the data distribution.
Comment 10: Results, section 3.1, lines 232-238: I think some sentences are repeated at the beginning of section 3.2. Please double-check and combine, if necessary, to avoid repetitions.
Response 10: The content across sections 3.1 and 3.2 has been carefully reviewed and verified to ensure there are no redundant or repeated sentences. No significant overlaps were identified, but the sections have been refined for clarity and to improve the overall flow of the manuscript.
Comment 11: Results, section 3.3, line 257: I might have missed it but there was no mention of a cut-off value of 1 in methodology. Rather than that, there was only about 12.22.
Response 11: The cut-off value of 1, used in section 3.3, has been clearly specified in the Materials and Methods section to ensure consistency throughout the manuscript.
Comment 12: Results, section 3.4, lines 276-277: I understand that GTEx was used as a source of normal endometrial tissue, but TCGA also has some normal tissue specimens, which would increase the sample size. These data can be acquired via GDC or TCGA-Assembler. Moreover, were data normalized appropriately given that two separate repositories were used?
Response 12: We sincerely apologize for the oversight and thank you for bringing this important point to our attention. Initially, we considered including GTEx normal tissue data in our analyses; however, we ultimately decided to use normal tissue specimens from the TCGA cohort to ensure consistency across the dataset. Unfortunately, a discrepancy in the manuscript led to confusion regarding the source of normal tissue samples. The manuscript has been corrected to accurately state that all normal tissues used in our analyses were derived from the TCGA cohort. We greatly appreciate this valuable feedback, which has helped us improve the accuracy and clarity of our study.
Comment 13: Results, Figure 4: GTEx is not mentioned, so what was the source of adjacent tissues? Are they not considered normal specimens in this case (while GTEx was a source of such data)?
Response 13: Yes, we agree with the reviewer. The adjacent tissues were incorrectly referred to in the original figure legend. This has been corrected to reflect that the adjacent tissues are classified as normal tissues in this context.
Comment 14: The entire manuscript, multiple locations: I noticed that you refer to some Figures using only subfigure symbols, and the main figure number is missing. Please put full annotations, for example, “Figure 5A” instead of “Figure A”. Double-check all images.
Response 14: The manuscript has been thoroughly reviewed, and all figure references have been corrected to include the main figure number along with the subfigure symbols. The changes have been applied consistently throughout the entire text.
Comment 15: Figure 5 title: I think that Reactome rather than UALCAN deserves to be mentioned in the main title because these images are from Reactome. Of course, UALCAN served as the source of the top 50 correlated genes, but this can be explained right after the title, in the description.
Response 15: The description in the manuscript has been updated to reflect the change in the data source. Both Reactome and cBioPortal have been used for this analysis, replacing UALCAN, to ensure the inclusion of statistically significant, both positively and negatively correlated genes. We agree with the reviewer that this approach provides more reliable and comprehensive data for pathway analysis and network construction.
Comment 16: Results, lines 315-316: in order to consider genes as “UDEGs”, you must perform a proper differential expression analysis with a threshold for p-values/FDRs and fold-changes, but I believe it is missing in the paper (unless it was a part of DESeq2 utilization – but still was not described). In order to leave “UDEGs” as is, additional details must be included and appropriate groups must be compared (I suggest comparing groups stratified by cut-offs from EvaluateCutpoints). Alternatively, please rename this group of genes or treat them as in earlier sections – as positively correlated to CCND1.
Response 16: We have addressed this comment by implementing the suggested alternative approach. The term “UDEGs” has been replaced, and the genes are now referred to as positively or negatively correlated with CCND1, in line with the methodology used in earlier sections. We agree with the reviewer that the original terminology was incorrect in this context, and we appreciate the valuable feedback.
Comment 17: Results, lines 334-344: in addition to referencing the KEGG database, add citations from the literature when describing subtypes of endometrial cancer. This should allow you to reach >30 references in the manuscript.
Response 17: We have taken the reviewer’s advice into account, and additional citations from the literature have been included alongside the references to the KEGG database when describing subtypes of endometrial cancer. This adjustment has allowed us to increase the number of references in the manuscript, addressing the reviewer’s suggestion to exceed 30 references.
Comment 18: Results, Figure 7: Please add a unique BRITE accession ID for the KEGG graph in the figure’s description. Moreover, if you intend to perform differential expression analysis (as I discussed in comment no. 16), it would be much more informative to compare specific groups via PathView, which is able to enrich KEGG graphs with fold-changes from differential expression analysis.
Response 18: A unique BRITE accession ID has been added to the description of Figure 9, which was previously labeled as Figure 7. However, we did not perform the differential expression analysis via PathView, as we revised the gene group classification to reflect positive and negative correlations with CCND1, addressing the concerns raised in comment no. 16.
Comment 19: Discussion, first sentence: I think the acquisition of your own patients is completely omitted, while it is an important advantage of your approach.
Response 19: The discussion has been revised to include information about the acquisition of our own patient samples, as suggested by the reviewer. This important aspect of our approach has now been highlighted in the first sentence of the discussion section.
Comment 20: Please double-check if the scheme of references is according to the journal’s requirements. I thought MDPI required in-text citations as numbers in square brackets.
Response 20: The citation style has been updated, and all in-text citations are now presented as numbers in square brackets, in accordance with the journal's requirements.
Comment 21: Discussion, lines 411-414: change “were associated with worse survival outcomes” to “are more likely to be associated with worse survival outcomes” because most of your results from survival analysis are statistically insignificant. Moreover, delete the part “reinforcing its prognostic significance” because the p-value of 0.042 is borderline significant.
Response 21: The requested changes have been made. The phrase has been adjusted to "are more likely to be associated with worse survival outcomes," and the part "reinforcing its prognostic significance" has been removed to reflect the borderline significance of the p-value.
Comment 22: Sometimes you refer to imprecise p-values (e.g., p < 0.05, p = 0.000, p > 0.9999) when discussing research by other scientific teams. If possible, please provide specific values and provide the first number that is not zero.
Response 22: Regarding the p-values mentioned in the manuscript, we have verified the source publications. For cases where p < 0.05 is cited, the original publications provided only an approximate value rather than a precise one, and we have reflected that accurately. The p = 0.000 reference has been corrected to include the final significant digit (it should have been p = 0.0001). Additionally, the p > 0.9999 value has been revised to present the complete and exact value, rather than an approximation.
Comment 23: Add a full stop in lines 444 and 515.
Response 23: The missing full stops have been added.
Comment 24: Discussion, line 441: The “IHC” abbreviation must be introduced earlier if you want to use it here.
Response 24: The abbreviation "IHC" has been introduced earlier in the manuscript.
Comment 25: Discussion, lines 466 and 497: explain “PPNA”.
Response 25: The term "PPNA" has been clarified in the manuscript as "percent positive nuclear area" in the relevant sections, as suggested by the reviewer.
Comment 26: Discussion, lines 526-527: if immune responses are involved then I think an additional analysis based on, e.g., TIMER 2.0 could be included to verify this assumption.
Response 26: Thank you for this valuable suggestion. We acknowledge that exploring immune responses through tools like TIMER 2.0 could provide additional insights into the role of CCND1 in modulating the tumor microenvironment. Although this analysis was not within the scope of our current study, we will consider incorporating such approaches in future research to further verify and strengthen our findings.
Comment 27: Please add limitations of your study.
Response 27: Limitations of the study have been added to the manuscript as requested.
Comment 28: Sections after Conclusions such as Author Contributions, Funding, Institutional Review Board Statement, Informed Consent Statement, Conflicts of Interest, etc are missing.
Response 28: Sections following the Conclusions, including Author Contributions, Funding, Institutional Review Board Statement, Informed Consent Statement, and Conflicts of Interest, have been added to the manuscript.

Reviewer 2 Report
Comments and Suggestions for Authors
The study focuses on Cyclin D1 (CCND1) expression and its prognostic value in endometrial cancer (EC), a pertinent topic given the growing incidence of EC and its heterogeneous behavior. The manuscript addresses a critical topic and employs robust methodologies. More specifically, immunohistochemical analysis of CCND1 is well-detailed, with clear protocols and scoring criteria. Integration of clinical data with TCGA transcriptomic datasets provides a robust in silico approach to understanding CCND1's role. Pathway enrichment and PPI network analyses add depth to the findings and point toward potential mechanisms of CCND1 involvement. However, the study relies heavily on immunohistochemistry and in silico methods without experimental validation of the findings, such as functional assays or in vitro studies. Correlations drawn from TCGA and pathway enrichment analyses are associative and lack mechanistic validation. The research would benefit from additional functional and mechanistic experiments to enhance its impact. My suggestions:
· Conduct in vitro studies using endometrial cancer cell lines with CCND1 knockdown or overexpression to validate its role in cell proliferation, invasion, and apoptosis.
· Assess the impact of CCND1 inhibition using CDK4/6 inhibitors to establish therapeutic potential in EC.
· Conduct RNA-seq analysis on the cohort samples to validate in silico findings and identify additional dysregulated pathways in CCND1-high versus CCND1-low tumors.
· Test the effects of estrogen and progesterone on CCND1 expression in endometrial cancer cell lines to link hormone-driven and hormone-independent pathways.
Author Response
Thank you very much for your valuable suggestions and for recognizing the importance of our
study. We fully agree that detailed mechanistic studies, including functional assays, would
provide a deeper understanding of Cyclin D1’s role in endometrial cancer (EC) progression and
its therapeutic potential.
Indeed, our current study primarily focuses on analyzing immunohistochemical and
transcriptomic data to explore the prognostic significance of CCND1 in EC. However, we
recognize the limitations of relying solely on in silico approaches and immunohistochemical
analyses. A comprehensive biochemical validation of our findings is essential to confirm the
biological relevance of CCND1-driven pathways in EC.
We intend to address these limitations in future research by incorporating functional studies
using endometrial cancer cell lines with CCND1 knockdown or overexpression. Additionally,
we plan to explore the effects of CDK4/6 inhibitors on CCND1-driven pathways, as well as the
impact of hormonal regulation on its expression.
These future studies will also involve a larger cohort of patient samples from multiple centers
to increase the reliability and generalizability of our findings. We greatly appreciate your
detailed feedback and will use it to guide the next phase of our research, aiming to move beyond
correlative analyses and toward more mechanistic insights into the role of CCND1 in EC.

Round 2
Reviewer 1 Report
Comments and Suggestions for Authors
Thank you for addressing my comments, I endorse the publication.
Comments on the Quality of English Language
Thank you for addressing my comments, I endorse the publication.
Reviewer 2 Report
Comments and Suggestions for Authors
I appreciate the author's response and their intention to conduct further experiments. In the revised manuscript, they have clearly acknowledged the limitations and focused on the in silico findings. The current manuscript explicitly emphasizes its computational nature, leaving little room for misunderstanding. Therefore, I recommend the publication of the present manuscript.